# Monitoring NLRP3 Inflammasome Activation and Exhaustion in Clinical Samples: A Refined Flow Cytometry Protocol for ASC Speck Formation Measurement Directly in Whole Blood after Ex Vivo Stimulation

**DOI:** 10.3390/cells11203306

**Published:** 2022-10-20

**Authors:** Rémy Coudereau, Morgane Gossez, Bénédicte F Py, Thomas Henry, Anne-Claire Lukaszewicz, Guillaume Monneret, Fabienne Venet

**Affiliations:** 1Immunology Laboratory, Edouard Herriot Hospital, Hospices Civils de Lyon, 5 Place d’Arsonval, CEDEX 03, 69437 Lyon, France; 2EA 7426 “Pathophysiology of Injury-Induced Immunosuppression” (Université Claude Bernard Lyon 1—Hospices Civils de Lyon—bioMérieux), Joint Research Unit HCL-bioMérieux, 69003 Lyon, France; 3CIRI, Centre International de Recherche en Infectiologie, Univ Lyon, Inserm U1111, Université Claude Bernard-Lyon 1, CNRS, UMR5308, ENS de Lyon, 69007 Lyon, France; 4Anaesthesia and Critical Care Medicine Department, Edouard Herriot Hospital, Hospices Civils de Lyon, 5 Place d’Arsonval, CEDEX 03, 69437 Lyon, France

**Keywords:** NLRP3 inflammasome, ASC speck, flow cytometry, protocol, whole blood

## Abstract

Alteration of NLRP3 inflammasome pathway including hyper-activation or exhaustion has been implicated in the pathophysiology of many diseases. Following cell stimulation, aggregation of the ASC protein into a multiprotein complex, the ASC speck, has been proposed as a specific read-out for monitoring NLRP3 inflammasome activation by flow cytometry in clinical samples. So far, only a few papers have described a technique to detect ASC speck formation directly in whole blood without any cell purification, and none included an ex vivo stimulation. The objective of this study was thus to develop a simple and shortened flow cytometry protocol to detect ASC speck formation directly in whole blood including an ex vivo stimulation step. We showed that after red blood cells lysis and removal of the LPS stimulation step, ASC speck formation can be detected in both monocytes and neutrophils from healthy donors directly in nigericin-stimulated whole blood samples. Using samples from four septic shock patients, we showed that this technique allows for the detection of NLRP3 inflammasome exhaustion in clinical samples. This novel shortened and simple whole blood protocol should facilitate day-to-day monitoring of NLRP3 inflammasome activation and exhaustion in both monocytes and neutrophils in clinical studies.

## 1. Introduction

Inflammasomes are multiprotein platforms involved in innate immune response. They are activated following danger signal recognition in a two-step process called “priming” and “activation”, leading to the cleavage of the pro-caspase-1 in activated caspase-1. NLRP3 is the most studied canonical inflammasome. It is composed of a sensor (NLRP3), recruiting a central adaptor protein known as ASC (apoptosis-associated speck-like protein containing a caspase activating and recruitment domain), which polymerizes after activation to a supramolecular signaling platform called the ASC speck. This leads to the recruitment and activation of caspase-1, responsible for pro-IL1-β and pro-IL-18 cleavages into their pro-inflammatory active forms [1].

Aberrant activation of NLRP3 inflammasome has been implicated in the pathophysiology of many inflammatory diseases such as auto-inflammatory periodic fever syndromes, including cryopyrin-associated periodic syndromes [2] and familial Mediterranean fever [3] but also in cancer, diabetes, gout, cardiovascular [4] and neurological disorders [5]. Conversely, altered NLRP3 inflammasome activation capacity called NLRP3 inflammasome exhaustion has been described in infectious diseases such as sepsis [6] or severe COVID-19 [7] in association with deleterious outcomes. This highlights the necessity to develop robust immunomonitoring assays to evaluate NLRP3 inflammation activation and exhaustion in clinical samples.

Monitoring of NRLP3 inflammasome activation in human samples has mainly been performed through detection of IL1-β, IL-18 secretion or caspase-1 activation after cell stimulation ex vivo. However, these techniques require lengthy cell purification and/or stimulation steps, which may have limited their use in clinical studies. More recently, it has been shown that NLRP3 activation can be monitored through the detection of the intracytoplasmic polymerization of the ASC protein (i.e., speck formation) [8], which might represent a more specific read-out of NLRP3 inflammasome activation than cytokine secretion measurement [9]. Indeed, before activation, a weak and diffuse ASC fluorescence is observed in cells. After activation, ASC aggregates into a single concentrated bright fluorescent speck in the perinuclear region, which can be detected by flow cytometry, while the rest of the cell is mostly depleted of any fluorescence signal [10].

So far, only a few papers have described a technique to detect ASC speck formation directly in whole blood without any cell purification. In addition, to the best of our knowledge, none of these protocols included an ex vivo stimulation. This is very important as such stimulation step will allow for the detection of NLRP3 inflammasome pathway exhaustion in clinical studies in addition to the monitoring of basal NLRP3 inflammasome pathway activation in vivo.

In this context, the objective of the current work was to define a simple flow cytometry protocol for the measurement of ASC speck formation directly in human whole blood including an ex vivo stimulation step. Such refined technique will help to simultaneously assess basal NLRP3 pathway activation level in vivo and its inhibition or exhaustion following re-activation ex vivo in clinical studies.

## 2. Materials and Methods

### 2.1. Reagents

The following antibodies were used: CD45-PB, CD14-APC (both from Beckman Coulter, Brea, CA, USA, clones J33 and RMO52, respectively), CD15-AF700 and anti-ASC-PE (both from Biolegend, San Diego, CA, USA, clones W6D3 and HASC-71, respectively). Following reagents were used: phosphate-buffered saline (PBS) (Eurobio Scientific, Luxembourg, Luxembourg), Ficoll (Dutscher, Bruxelles, Belgium), RPMI HEPES (Eurobio, Les Ulis, France), penicilline/streptomycine (Clinisciences, Nanterre, France), fungizone (Cheplapharm Arzneimittel, Mesekenhagen, Germany), glutamine (Ozyme, Saint-Cyr-l’École, France), versalyse (Beckman Coulter), lipopolysaccharide (LPS) (Sigma-Aldrich, single mix of references L3137, L2637 and L3012, Saint-Louis, MO, USA), nigericin (InvivoGen, San Diego, CA, USA), water for injection (Aguettant, Lyon, France), Cytofix/Cytoperm, Fixation/Permeabilization Kit, stain buffer (both from BD Bioscience, Franklin Lakes, NJ, USA) and human AB serum (SAB) (“Etablissement Français du Sang”, Lyon, France).

### 2.2. Blood Sample Processing

Fresh peripheral blood from healthy donors (HD) was provided by the Etablissement Français du Sang (French National Blood Bank) from Lyon using heparin anticoagulant tubes. A written non-opposition to the use of donated blood for research purposes was obtained from the HD. PBMCs were isolated using density gradient centrifugation. Briefly, heparin blood was diluted 1:1 with PBS, carefully pipetted on the ficoll solution and centrifuged for 20 min at 770× *g* with low acceleration and low brake applied. The PBMCs ring was removed by pipetting and the cells washed with PBS by centrifugation for 10 min at 800× *g*, 3 times. After the final wash, cells were re-suspended in 1 mL of RPMI HEPES + penicilline/streptomycine (20 µg/mL) + fungizone (2.5 µg/mL + glutamine (2 mM). In addition, 4 septic shock patients, with blood drawn during the first 48 h after ICU admission, were tested to illustrate the potential of such whole blood protocol in clinical samples. Patients were included in the IMMUNOSEPSIS-4 cohort. Diagnostic criteria for septic shock was based on the Sepsis-3 definition [11]. Exclusion criteria disqualified patients under 18 years of age, subjects with aplasia or pre-existent immunosuppression, pregnant women, and institutionalized patients. This project was approved by Institutional Review Board for ethics (“Comité de Protection des Personnes Ouest II”, n° RCB: 2019-A00210-57). This study is registered at the French Ministry of Research and Teaching (#DC-2008-509), at the Commission Nationale de l’Informatique et des Libertés, and on https://clinicaltrials.gov (#NCT04067674).

### 2.3. ASC Staining in PBMCs

In this stage, 2.5 million cells/mL in RPMI HEPES were incubated with or without LPS at 0.1 µg/mL for 3 h, at 37 °C, followed by incubation with or without nigericin at 10 µg/mL for additional 30 min, at 37 °C. Cells were washed with 1 mL of stain buffer, centrifuged for 6 min at 300× *g* and supernatant removed. PBMCs were then labelled with 5 µL of PB-labelled anti-CD45 antibody and 5 µL of APC-labelled anti-CD14 antibody, incubated for 15 min, at room temperature, in the dark and washed with 1 mL of stain buffer for 6 min at 300× *g*. Permeabilization was performed by using 100 µL of Cytoperm reagent, incubated for 20 min, at 4 °C. After two washing steps with 1 mL of 1:10 diluted Perm/Wash buffer, cells were incubated with PE-labelled anti-ASC antibody for 20 min, at room temperature, in the dark. A final washing was performed with diluted Perm/Wash buffer, supernatant removed, and cells re-suspended in 300 µL of PBS. Samples were run on a BD FACS ARIA II (BD Bioscience, Franklin Lakes, NJ, USA) and results expressed as percentage of ASC speck-positive cells among respective cell populations.

### 2.4. Cell Sorting and Confocal Microscopy

In stimulated and unstimulated conditions, monocytes were sorted on a BD FACS ARIA II cytometer (BD Bioscience). Among PBMCs, monocytes were first selected based on CD14 expression. Cells were then sorted based on ASC staining profile with a homogeneous ASC-staining profile of unstimulated monocytes and a bi-modal ASC staining in monocytes after LPS and nigericin stimulations. Sorted cells were centrifuged for 6 min at 2200× *g*, supernatant removed and cells re-suspended in 200 µL of SAB. Finally, cells were fixed on a glass slide using a cytospin for 5 min at 3000 rpm. Cells were imaged using Confocal Zeiss LSM980 (Carl Zeiss, Marly le Roi, France) and 10x objectives. Images were analyzed using ZEN lite software (version 3.4, Carl Zeiss, Marly le Roi, France).

### 2.5. ASC Staining in Whole Blood

In order to remove red blood cells, 100 µL of whole heparin blood was first diluted in 1 mL of lysis solution (Versalyse) for 10 min, at room temperature, in the dark. Cells were then washed with 1 mL of PBS for 10 min, at room temperature, in the dark and centrifuged for 6 min at 300× *g*. After supernatant removal, cells were re-suspended in 500 µL of PBS and incubated with or without LPS at 0.1 µg/mL for 3 h, at 37 °C, then with or without nigericin at 10 µg/mL for 30 min, at 37 °C. Cells were washed with 1 mL of stain buffer, centrifuged for 6 min at 300× *g* and supernatant removed. Cells were then labelled with 5 µL of PB-labelled anti CD45-antibody, 5 µL of APC-labelled anti-CD14 antibody and 5 µL of AF700-labelled anti-CD15 antibody, incubated 15 min, at room temperature, in the dark and washed with 1 mL of stain buffer for 6 min at 300× *g*. A permeabilization step was performed by using 250 µL of Cytoperm, incubated for 20 min, at 4 °C. After two washings steps with 1 mL of 1:10 diluted Perm/Wash buffer, cells were labelled with 5 µL of PE-labelled anti-ASC antibody for 20 min, at room temperature, in the dark. A final wash was performed with 1 mL of diluted Perm/Wash buffer, supernatant removed and cells re-suspended in 300 µL of PBS. Samples were run on a FACS ARIA II cytometer and results expressed as percentage of ASC speck-positive cells among respective cell population.

### 2.6. Statistical Analysis

Results were presented as individual values and boxplots. Statistical analysis were performed using the non-parametric Wilcoxon paired test. Data were analyzed using R Studio software (version 1.2.5001; R studio, Boston, MA, USA).

## 3. Results

### 3.1. Flow Cytometry Measurement of ASC Speck in PBMCs

Gating strategy is presented in Figure 1A. Cells were first selected based on side scatter (SSC)/forward scatter (FSC) characteristics and doublets were then excluded using FSC-area/FSC-height histogram. Leukocytes were identified based on CD45 expression. Monocytes were selected based on CD14 expression among CD45+ leukocytes. Finally, monocytes were analyzed for ASC–width (ASC-W) and ASC-area (ASC-A) signals. Without stimulation, monocytes showed a homogenous population presenting high ASC-W and high ASC-A. After stimulation, cells with ASC speck formation could be detected based on a diminished ASC-W signal. Results obtained in six HD are depicted in Figure 1B. Without stimulation or in case of stimulation with either LPS or nigericin alone, percentages of ASC speck-positive monocytes remained low (i.e., 0.06%, 0.85% and 0.44%, respectively (Figure 1B)). In sharp contrast, the percentage of ASC speck-positive monocytes increased significantly following stimulation with both LPS and nigericin (median = 22.7%, *p* < 0.05 vs. other conditions).

### 3.2. Microscopic Detection of ASC Speck in PBMCs

To confirm that the modification of ASC staining profile observed in monocytes after stimulation corresponded to ASC speck formation in these cells, unstimulated resting monocytes, and both ASC speck-positive and ASC speck-negative monocytes after LPS and nigericin stimulations, were purified (Figure 2 left panels) to be analyzed by confocal microscopy. By microscopy, purified monocytes are presented with clear CD45 (blue) and CD14 (red) expressions. As expected (Figure 2 right panels), a diffuse ASC fluorescence (green) was observed in sorted non-stimulated monocytes and ASC speck-negative monocytes after stimulations. In contrast, in ASC speck-positive monocytes, ASC concentrated into a single bright fluorescent speck in the perinuclear region, depleting the rest of the cell of the fluorescence signal. This confirmed the reliability of flow cytometry analysis to specifically distinguish ASC speck-positive monocytes.

### 3.3. Measurement of ASC Speck-Positive Monocytes in Whole Blood by Flow Cytometry

We next attempted to transfer this protocol to whole blood samples. Thus, we added a lysis step to remove red blood cells prior to stimulation and staining (as previously described). A representative example of a flow cytometry staining in one healthy donor is shown in Figure 3A, and the results obtained in six HD are presented in Figure 3C (left panel). Similarly to previous observation in PBMCs, no monocytes with speck were detected in unstimulated condition or after stimulation with LPS alone. Unexpectedly, unlike previous results in PBMCs, stimulation with nigericin alone led to elevated percentage of ASC speck-positive monocytes (medians = 60.6% vs. 0.44% in PBMCs). Moreover, stimulation with both LPS and nigericin strongly increased percentages of monocytes with speck compared with results observed in PBMCs (medians = 69.4% vs. 22.7%, respectively).

We postulated that red blood cell lysis was responsible for monocyte priming so that activation with nigericin alone was sufficient to induce ASC speck formation in monocytes. To confirm this hypothesis, PBMCs from 2 HD were treated or not with red blood cell lysis buffer prior to nigericin stimulation (Figure 3B). In untreated samples, we confirmed that no ASC speck formation was detected in samples treated with nigericin alone, while a good response was measured in samples treated with LPS + nigericin (Figure 3B, right panel). In contrast, in PBMCs treated with cell lysis buffer, nigericin stimulation alone led to an increased percentages of monocytes forming speck equivalent to the one observed in cells stimulated with LPS and nigericin (Figure 3B, left panel). This confirmed that in this experimental setting, red blood cell lysis was sufficient to induce cell priming of NLRP3 inflammasome pathway.

Finally, we assessed the ASC speck formation in samples from four septic shock patients sampled within the first 48 h after ICU admission (Figure 3C right panel). These patients were elderly male (median age = 64 years) and presented with high severity scores (median of Sequential Organ Failure Assessment score (SOFA) = 10.5 and median of Simplified Acute Physiology Score II (SAPS II) = 50.5). These preliminary results showed a decreased percentage of ASC speck-positive monocytes (median = 38%) following stimulation with nigericin alone compared to HD.

Thus we have set-up a whole blood flow cytometry protocol to monitor ASC speck formation in monocytes usable in clinical samples.

### 3.4. Measurement of ASC Speck-Positive Polymorphonuclear Neutrophils (PMN) in Whole Blood by Flow Cytometry

In contrast to PBMCs, the use of a whole blood flow cytometry protocol enables the study of ASC speck formation in PMN; thus, we evaluated ASC speck formation in neutrophils using the current technique. The initial gating strategy for PMN identification was similar to monocytes (Figure 4A). PMN were specifically selected on their positive CD15 expression. As for monocytes, ASC speck-positive cells were identified based on their diminished ASC-width signal. In six HD, ASC speck formation in non-stimulated PMN was low (Figure 4B, median = 0.01). Upon stimulation by nigericin alone or LPS and nigericin, the percentage of ASC speck-positive PMN increased significantly (Figure 4B). However, the percentage of ASC speck-positive PMN remained low after stimulation in comparison with results observed in monocytes (Figure 4B).

Finally, preliminary results obtained in septic shock patients did not show any decrease in the percentage of ASC speck-positive PMN after stimulation compared with HD (data not shown).

## 4. Discussion

NLRP3 inflammasome is implicated in a number of highly prevalent clinical conditions and has been consequently largely investigated by different techniques both in experimental set-up and at clinical levels. A first approach consists of detecting inflammasomes activation through cytokines production (namely IL-1β and IL-18) commonly detected by ELISA or immunoblot analysis in plasma or after ex vivo stimulation of purified cells. However, IL-1β cleavage and release may occur through inflammasome-independent processes [12]. Thus, this approach may not be specific to inflammasomes. An additional method is based on the detection of caspase 1 activation through immunoblot of purified cells or by flow cytometry in whole blood [13]. This latest technology presents the benefit of studying the caspase 1 activation in specific cell types and subpopulations and has been used to investigate the NLRP3 inflammasome activation in COVID-19 patients [13]. However, as for cytokine release assays, caspase 1 activation may not be fully specific of inflammasome activation but may be confounded with other non-inflammasome-related caspase or protease activities [9]. Another way of studying inflammasomes activation relies on cell death detection, but the distinction of pyroptosis (specific to the inflammasome related cell death) from other forms of cell death remains challenging [14].

Overall, direct assessment of the ASC polymerization at the cellular level appears as the most specific technique to monitor inflammasomes activation. In addition, the use of nigericin stimulation provides a specific activation for NLRP3 inflammasome pathway among other inflammasomes. ASC speck formation could be assessed by microscopy [15]. However, this approach is not suitable for large-scale human studies. To circumvent microscopy practical drawbacks in clinical research, Sester and colleagues took advantage of flow cytometry to assess ASC speck polymerization in PBMCs following stimulation ex vivo [8]. With this approach, Martínez-García et al. reported the profoundly altered NLRP3 activation in septic patients in association with mortality [6]. Recently, after evaluation of various pre-analytical conditions (type of anticoagulant, temperature and sample storage time, etc.), Wittman and his team provided technical recommendations regarding ASC specks detection in human PBMCs [16]. In order to facilitate clinical explorations, Cui et al. reported a whole blood protocol without any subsequent ex vivo stimulation to study in vivo NLRP3 inflammasome activation level [17]. Consequently, the objective of the present study was to set up a refined whole blood protocol for ASC speck formation monitoring in primary human cells including an ex vivo stimulation step to follow both in vivo NLRP3 inflammasome activation level and ex vivo re-activation capacity. To the best of our knowledge, no study has, so far, described such a technique.

First, we demonstrated that the characteristic signal of ASC speck formation by flow cytometry (i.e., diminished fluorescence width) observed in PBMCs was also observed in whole blood. By using confocal microscopy on flow cytometry-sorted cells, we confirmed that cells identified by diminished ASC-width by flow cytometry show ASC speck. Based on this, we were able to monitor NLRP3 activation in whole blood from both HD and septic patients. In line with data reported by Martínez-García et al., our preliminary results showed a marked decrease in ASC speck formation in cells from patients suffering sepsis when sampled within the first 48 h after ICU admission.

In whole blood assay, high amount of ASC speck-positive monocytes were detected after LPS and nigericin stimulation as expected. Surprisingly, a similar level of ASC speck-positive monocytes was detected when stimulated with nigericin alone. Although the priming step of NLRP3 activation for cytokine release has been demonstrated to be sometimes dispensable [18,19]; priming (with LPS) preceding activation (with ATP or nigericin) considerably increases the activation of the inflammasome. It is noteworthy that we added a lysis pre-treatment step in the whole blood assay to remove red blood cells. We confirmed experimentally by using lysis buffer prior to NLRP3 inflammasome stimulation of PBMCs that cell stress caused by the initial use of lysis solution or that damage-associated molecular pattern molecules (DAMPs) released from lysed red blood cells could provide the priming signal. As a result, stimulation with nigericin alone following lysis buffer treatment enabled us to detect an increased percentages of monocytes with speck compared with results in PBMCs without lysis buffer treatment. Such a removal of the LPS stimulation step resulted in an easier and time-saving protocol.

Another asset of the described protocol is to allow the exploration of neutrophil present in whole blood but discarded upon the PBMCs purification step. Although inflammasomes were discovered and mainly studied in monocytes and macrophages, recent studies suggest that the importance of the neutrophil inflammasome has been underestimated [20]. Here, as observed in monocytes, we noticed an increase in ASC speck-positive neutrophils after stimulation. To note, although percentages of ASC speck-positive neutrophils were much lower than in monocytes upon stimulation; ASC speck formation in these cells may nevertheless be clinically relevant considering the high number of circulating neutrophils in healthy donors and their increased number in certain clinical conditions such as sepsis. In addition, these results are consistent with the study of Boucher et al., in which neutrophils assembled less ASC speck than macrophages [21]. Thus, the current protocol should allow for the robust monitoring of ASC speck formation in neutrophils in clinical samples.

## 5. Conclusions

We developed a whole blood-based flow cytometry protocol to detect baseline ASC speck formation downstream of NLRP3 inflammasome activation and NLRP3 re-activation abilities both in monocytes and neutrophils usable in clinical samples. The removal of two time-consuming steps (i.e., ficoll purification and LPS stimulation) shortened the protocol duration by several hours, and the addition of an ex vivo stimulation step allows for the monitoring of NLRP3 inflammasome exhaustion in clinical samples. Overall, this technique should permit an easier implementation of NLRP3 studies in clinical research protocols.

## Figures and Tables

**Figure 1 cells-11-03306-f001:**
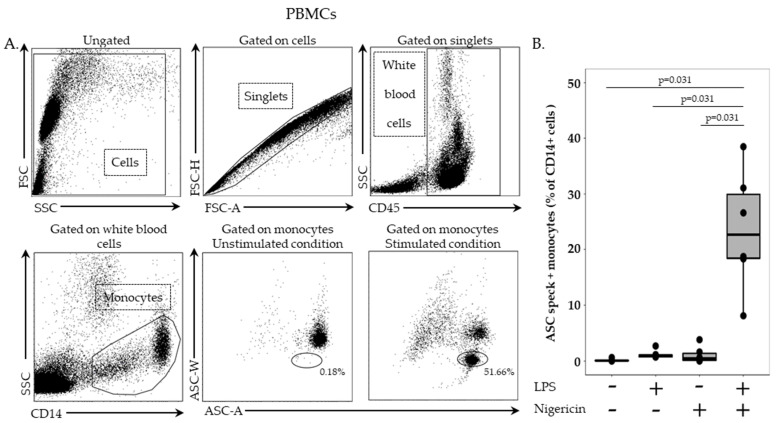
Monitoring of ASC speck-positive monocytes in PBMCs. (**A**) Gating strategy. After doublet exclusion, monocytes were identified based on CD45 and CD14 expressions and speck formation was assessed on a bi-parametric ASC-area (ASC-A)/ASC-width (ASC-W) histogram gated on selected cells. ASC speck-positive cells are presented with diminished ASC-width (circle) in comparison with non-specking cells. Results obtained in one representative healthy donor PBMCs without stimulation or treated for 3 h with 0.1 µg/mL LPS prior to inflammasome induction using 10 µg/mL nigericin for 30 min are presented. (**B**) Percentage of ASC speck-positive monocytes from 6 healthy donors are shown. Results of unstimulated PBMCs or after PBMCs incubation with or without LPS at 0.1 µg/mL for 3 h, at 37 °C, followed by incubation with or without nigericin at 10 µg/mL for additional 30 min, at 37 °C, are shown. Results were expressed as percentage of ASC speck-positive cells among CD14+ monocytes and are presented as individual values and box-plot. Statistical analysis were performed using the non-parametric Wilcoxon paired test.

**Figure 2 cells-11-03306-f002:**
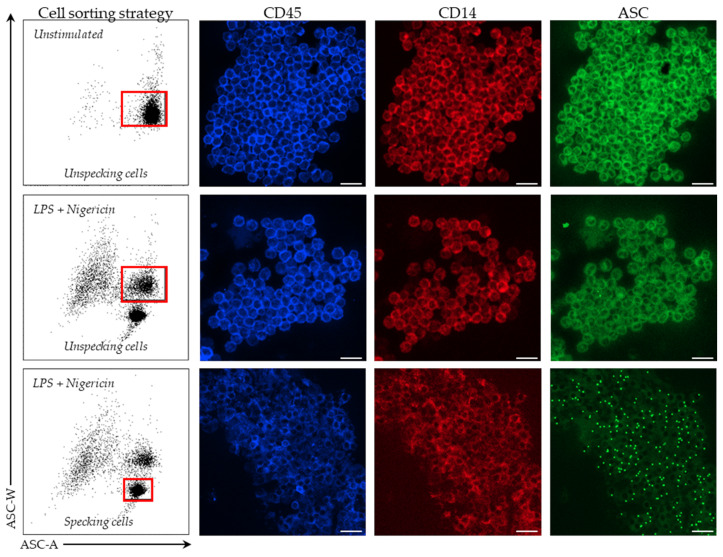
ASC speck detection by confocal microscopy in sorted monocytes. Fluorescence confocal microscopic images of ASC speck formation in PBMCs purified from 1 healthy donor after flow cytometry cell sorting in unstimulated situation or after 3 h stimulation with 0.1 µg/mL LPS prior to inflammasome induction using 10 µg/mL nigericin for 30 min are shown. Respective sorted populations are identified by red squares. Cells were stained for CD45 (blue), CD14 (red) and ASC (green). Monocytes were identified on the basis of CD45 (blue) and CD14 (red) expressions. Scale bar, 40 µm.

**Figure 3 cells-11-03306-f003:**
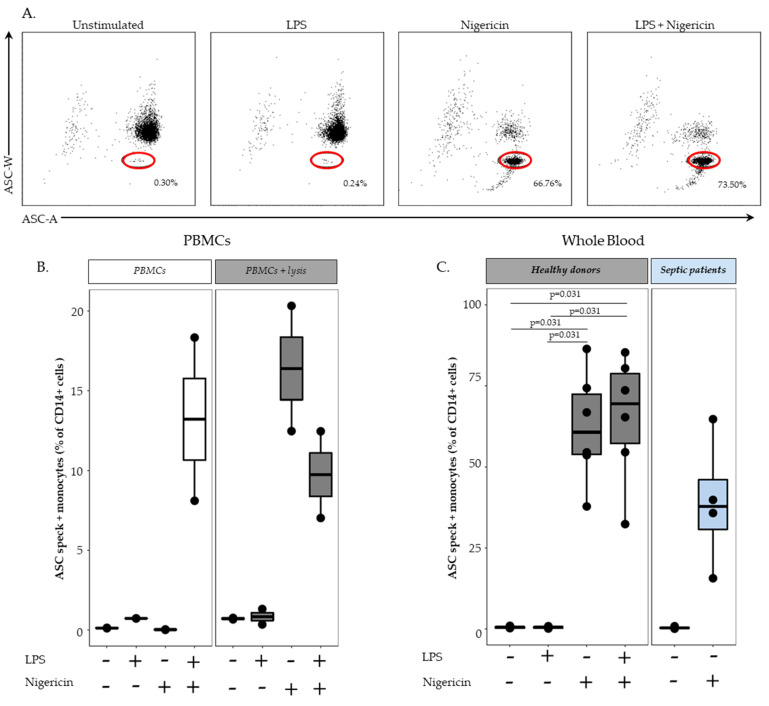
Monitoring of ASC speck-positive monocytes in whole blood and in PBMCs. (**A**) Detection of ASC speck-positive monocytes. Using a similar gating strategy as described in Figure 1 for PBMCs, speck formation was assessed in whole blood. Example of a bi-parametric ASC-area (ASC-A)/ASC-width (ASC-W) histogram in one representative healthy donor is shown. Activation conditions are indicated above each panel. Briefly, 100 µL of whole blood was treated with or without LPS at 0.1 µg/mL for 3 h, at 37 °C, followed by incubation with or without nigericin at 10 µg/mL for additional 30 min, at 37 °C, or left unstimulated. The percentages of ASC speck-positive cells among CD14+ monocytes (identified by red circles) in each condition are shown. (**B**) ASC speck formation upon stimulation in healthy donors. PBMCs from healthy donors (*n* = 2) were treated (**right** panel) or not (**left** panel) with lysis solution for 10 min, then with or without LPS at 0.1 µg/mL for 3 h, at 37 °C, followed by incubation with or without nigericin at 10 µg/mL for additional 30 min, at 37 °C, or left unstimulated. (**C**) ASC formation upon stimulation in healthy donors and septic patients. Here, 100 µL of whole blood from healthy donors (*n* = 6) was treated with or without LPS at 0.1 µg/mL for 3 h, at 37 °C, followed by incubation with or without nigericin at 10 µg/mL for additional 30 min, at 37 °C, or left unstimulated. Then, 100 µL of whole blood from septic patients *(n* = 4) within the first 48 h after ICU admission were treated for 30 min using 10 µg/mL nigericin or unstimulated. Results were expressed as percentage of ASC speck-positive cells among CD14+ monocytes and are presented as individual values and box-plot. Statistical analyses were performed using the non-parametric Wilcoxon paired test.

**Figure 4 cells-11-03306-f004:**
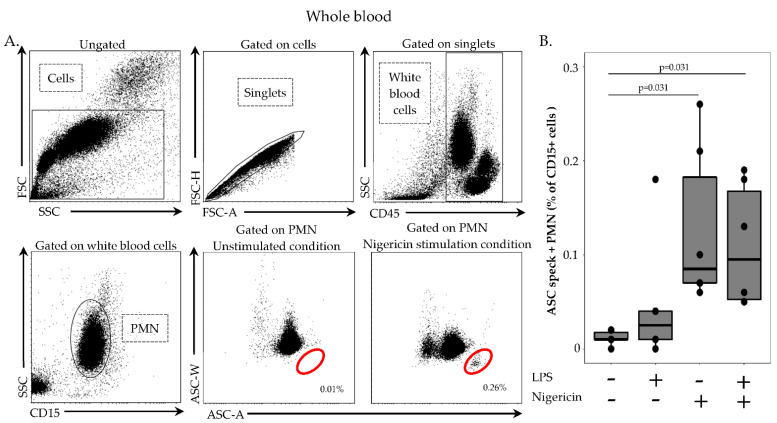
Monitoring of ASC speck-positive neutrophils in whole blood. (**A**) Detection of ASC speck-positive neutrophils. After doublet exclusion, neutrophils were identified based on CD45 and CD15 expressions. Then, ASC speck formation was assessed in selected cells based on a bi-parametric ASC-area (ASC-A)/ASC-width (ASC-W) histogram. Specking cells presented with diminished ASC-width (red circle) in comparison with non-specking cells. Representative flow cytometry staining in one healthy donor is shown. Activation conditions are indicated above each panel. Briefly, 100 µL of whole blood was treated with LPS at 0.1 µg/mL for 3 h, at 37 °C, followed by incubation with nigericin at 10 µg/mL for additional 30 min, at 37 °C, or left unstimulated. The percentages of ASC speck-positive cells among CD15+ neutrophils (identified by red circles) in each condition are shown. (**B**) ASC formation upon stimulation in healthy donors. Here, 100 µL of whole blood from healthy donors (*n* = 6) was treated with or without LPS at 0.1 µg/mL for 3 h, at 37 °C, followed by incubation with or without nigericin at 10 µg/mL for additional 30 min, at 37 °C, or left unstimulated. Results were expressed as percentage of ASC speck-positive cells among CD15+ PMN and are presented as individual values and box-plot. Statistical analysis were performed using the non-parametric Wilcoxon paired test.

## Data Availability

Data are available from corresponding author upon reasonable request.

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
