# Peer review of "Monitoring NLRP3 Inflammasome Activation and Exhaustion in Clinical Samples: A Refined Flow Cytometry Protocol for ASC Speck Formation Measurement Directly in Whole Blood after Ex Vivo Stimulation"

_cells, 2022, doi:10.3390/cells11203306_

Round 1
Reviewer 1 Report (Previous Reviewer 3)
Thank you for answering my questions and adapting the manuscript, which I hereby approve for publication. Congratulations to the authors!
Reviewer 2 Report (Previous Reviewer 1)
The authors have satisfactorily addressed all my comments.
This manuscript is a resubmission of an earlier submission. The following is a list of the peer review reports and author responses from that submission.
Round 1
Reviewer 1 Report
Comments on the manuscript # cells-1888779-v1 (Coudereau et al “Flow cytometry measurement of ASC speck formation in myeloid cells upon inflammasome activation: a whole blood protocol”).
The authors have developed a flow-based protocol for quantitation of inflammasome activation using whole blood. Although the approach has already been described in a number of previous publications, a refining could potentially help in clinical settings. The methodology description is enough detailed for reproduction by other labs, data presentation is clear and supports the conclusions. My opinion is “Acceptance with minor revision”.
Minor comments
11- The objective of this work is clearly defined in Introduction lines 68-71 and Discussion lines 341-342. Addition of the step of ex-vivo re-stimulation with nigericin, however, would be for detection of the NLRP3 inflammasome inhibition/exhaustion, rather than measurement of its activation in vivo. Please clearer address this in discussion and/or introduction.
22- Line 45. Abbreviations used only once (CAPS, FMF) are not necessary. Line 87 EFS=?
33- Line 48, typo “imp2licated”.
44- Lines 52, 58. “Productions” (of IL1-beta, IL-18) would be more precisely replaced with “secretion” or “maturation”.
55- Line 89 missing “the” (consents obtained from ALL donors).
66- Discussion, lines 305-310 redundant of introduction.
77- Line 351. Replace “As reported by…” with “In line with data reported by…”
Reviewer 2 Report
The authors present a study on the detection of ASC specks in human PBMCs from whole blood. Unfortunately, this technique has already been published in your Journal in November 2021, by Wittmann et al. The only difference between that study and the one presented here is the addition of 1) ASC speck detection in granulocytes by staining for CD15 prior to ASC and 2) the inclusion of ASC speck analysis in PBMCs from 4 sepsis patients.
There are some typos in the introduction. It was also unfortunate researchers could only access 4 x healthy donors for their standard speck assays and data is represented without any statistical analysis.
Reviewer 3 Report
In this protocol paper, Coudereau et al. describe a method to quantify ASC specks by flow cytometry in human PBMCs as well as in whole blood samples. Assessing inflammasome activation by measuring ASC specks in PBMCs was described already in the initial publication of this method by Sester et al. in 2015 (https://doi.org/10.4049/jimmunol.1401110, and recently Wittmann et al. published a very detailed protocol for ASC speck quantification in PBMCs in Cells (doi: 10.3390/cells10112880). Moreover, ASC speck quantification has also been described already on unstimulated cells from whole blood by Cui et al. in 2020 (doi: 10.3389/fimmu.2020.613745). As such, the novelty of the present protocol lies in the ex vivo stimulation of cells from whole blood, in which the authors show that nigericin suffices to induce ASC speck formation in whole blood monocytes. About 70% of HD whole blood monocytes exhibited ASC specks after nigericin stimulation (much more than in LPS/nigericin treated PBMCs), which the authors hypothesize to be a consequence of the red blood cell lysis step in their protocol. This high ASC speck response rate due to a technical requirement in the protocol represents a drawback for implementing the authors’ protocol in the clinic, as it may be difficult to use the protocol for diagnostically identifying patients with increased inflammasome activity. Conversely, the inability of the authors to convincingly show ASC speck responses of whole blood neutrophils may render also this aspect of the protocol difficult to implement in the clinic. Nevertheless, I appreciate the authors’ efforts trying to improve clinical practices to measure inflammasome activities and I therefore believe this protocol can be published after addressing the concerns below regarding differences with previously published protocols and potential improvements for the current protocol.
Major comments:
1. The authors show that only LPS/nigericin was capable of inducing ASC specks in PBMCs from HDs. However, Wittmann et al. previously showed that nigericin (in contrast to ATP) was sufficient to induce ASC specks in healthy PBMCs. How do the authors explain this difference? What is different in the protocols of both studies that could explain this?
2. The authors show that nigericin-treated whole blood monocytes of sepsis patients show less ASC specks than those of HDs. This is in accordance with the Nlrp3 immunosuppressive state that was previously demonstrated in PBMCs of sepsis patients (https://doi.org/10.1038/s41467-019-10626-x). However, this study showed that the Nlrp3 immunosuppressive state was transient, as PBMCs from advanced sepsis patients regained their ASC speck forming ability. Therefore, can the authors clarify at which stage of sepsis the donors of whole blood were? In addition, did the reduced ASC speck formation in whole blood monocytes from these patients correlate with ASC speck formation in PBMCs? Do also PBMCs from these sepsis patients display less ASC speck with the authors’ protocol?
3. The authors observed high levels of ASC speck formation in nigericin-treated whole blood monocytes, which may limit the use of this protocol in the clinic. Can the authors modify the protocol to obtain lower levels of ASC speck formation? For instance, can the authors try lower concentrations of nigericin? Does ATP also suffice to induce ASC specks in these whole blood monocytes?
4. The authors suggested that the ASC speck formation in nigericin-treated whole blood monocytes was due to priming during the RBC lysis step, either due to the lysis buffer or due to DAMPs released from lysed RBCs (lines 360-363). These are two options that can easily be investigated experimentally. Can the authors treat PBMCs with either the RBC lysis buffer or with lysed RBCs to evaluate which one can serve as the priming agent before nigericin activation of the inflammasome?
5. The ASC speck formation in nigericin- or LPS/nigericin-treated whole blood neutrophils is not very convincing due to the low levels of ASC specks and the variation between the whole blood donors. If the authors want to claim on lines 264-265 that ‘Upon stimulation by nigericin alone or LPS and nigericin, the percentage of ASC speck-positive PMN increased (figure 4b)’ they should perform statistical analyses to support this statement.
Minor comment
1. As the novelty of the paper lies in the ex vivo stimulation of whole blood I would suggest to change the title of the manuscript accordingly.